



# On the impact of Himalaya-induced gravity waves on the polar vortex, Rossby wave activity and ozone

Ales Kuchar[1], Petr Sacha[2,3], Roland Eichinger[2,4], Christoph Jacobi[1], Petr Pisoft[2], and Harald E. Rieder[3]

[1]Institute for Meteorology, Leipzig University, Stephanstr. 3, 04103 Leipzig, Germany
[2]Department of Atmospheric Physics, Faculty of Mathematics and Physics, Charles University, V Holesovickach 2, 180 00 Prague 8, Czech Republic
[3]Institute of Meteorology and Climatology, University of Natural Resources and Life Sciences, Vienna (BOKU), Gregor-Mendel-Strasse 33, 1180 Vienna, Austria
[4]Deutsches Zentrum für Luft- und Raumfahrt (DLR), Institut für Physik der Atmosphäre, Oberpfaffenhofen, Germany

**Correspondence:** Ales Kuchar (ales.kuchar@uni-leipzig.de)

**Abstract.** The instability of the Northern Hemisphere polar vortex is mainly caused by the breaking of planetary-scale (Rossby) waves (RWs). However, gravity waves (GWs) may also play an important role in polar vortex preconditioning before break-down events. Moreover, GWs affect dynamics in the stratosphere by altering the upward propagation of RWs at short time scales and therefore indirectly influence polar vortex stability. Due to the coarse spatial resolution of global chemistry-climate
models, current efforts in climate research rely on simulations where the majority of the GW spectrum is parameterized. In the present study, we apply a recently developed method for detecting strong orographic gravity wave (OGW) drag events in the lower stratosphere above the Himalayas. For this, we use a specified dynamics simulation of the chemistry-climate model CMAM (Canadian Middle Atmosphere Model) spanning the period 1979–2010. We show that strong OGW drag events above the Himalayas are associated with anomalously increased upward RW propagation in the stratosphere. This, in turn, is associ-
ated with an increase of the refractive index in the mid-latitude lower stratosphere, a region where the OGW drag dominates. Our results also illustrate that OGW strong events have the potential to alter ozone variability via changes of mixing in the surf zone and advection from lower latitudes. Altogether, we detail a preconditioning process of the polar vortex morphology by GWs above the Himalayas and how this relates to the proximity to polar vortex breakdown.

## 1 Introduction

The polar vortex is commonly associated with a westerly large-scale circulation around the pole in middle or high lati-
tudes (Waugh et al., 2017). Here, we refer to the polar vortex in the stratosphere, which plays a dominant role in the dynamical downward coupling to the troposphere during winter (Hardiman and Haynes, 2008). The coupling episodes are strongest during and after so called sudden stratospheric warmings (SSWs; Hitchcock and Simpson, 2014). SSWs are abrupt dynamical



events, during which polar stratospheric temperatures increase heavily, and the polar vortex is strongly weakened or completely reversed (Baldwin et al., 2021).

State-of-the-art long-range weather prediction systems use high-top model configurations to resolve processes in the stratosphere and even lower mesosphere. This has been motivated by the accumulated evidence of stratospheric variability affecting surface conditions in the mid-latitudes (see the recent review by Scaife et al., 2021). As mentioned above, a key factor of strato-

spheric influence on the troposphere is the polar vortex, its formation, strength and especially its breakdown. These extreme vortex events can be skillfully forecasted beyond the 10-day limit (Tripathi et al., 2015) but this is only possible for large spatial scales (Boer, 2003), while the predictability of subsynoptic scales is below this limit (Jung and Leutbecher, 2008). Lorenz (1969) first noted the unpredictability of small-scale eddies influencing larger spatial scales leading to an upscale transfer of uncertainty.

A vast body of literature exists on analyzing the dynamics before and after the SSW events, both in the stratosphere and in the troposphere (e.g. Albers and Birner, 2014; Jucker and Reichler, 2018; King et al., 2019). General consensus sees large-scale Rossby waves (RWs) as a key dynamical forcing influencing the vortex. Much less is known about the dynamical influence of small-scale internal gravity waves (GWs), which are ubiquitous in the winter atmosphere (Ern et al., 2018).

GWs exist on a variety of scales, and depending on the resolution of the model, a significant portion of their spectrum remains

unresolved in chemistry-climate models (CCMs) and also in global weather prediction models. This lack in the models is filled with GW parameterizations. In current generation of CCMs, two schemes are typically employed to parameterize dynamical effects of GWs resulting from orography (orographic gravity waves, OGWs) and other sources separately. A key output of GW parameterizations is the GW drag (GWD) — a wind and temperature tendency that is added to the CCM dynamics. Although CCMs have substantially increased in complexity, their GW parameterizations still rely on a diverse set of assumptions (e.g.,

strictly vertical instantaneous propagation) that are known not to occur in this form in the real atmosphere (Krisch et al., 2017). However, the assumptions are well justified to a leading order (Plougonven et al., 2020) and the schemes are tuned to help the models reproducing the zonal mean climatologies close to observations. Still, the details of the dynamical effects from the parameterized GWD are to a large extent unconstrained.

Our knowledge on GW effects, ranging from the regionality of precipitation (Cohen and Boos, 2017) to their importance for

the structure of atmospheric dynamics (e.g. Kim et al., 2003; Alexander, 2010; Eichinger et al., 2020), has so far been based on their predominantly parameterized effects. At altitudes between the upper troposphere and the mid stratosphere the dominant parameterized GWD component in CCMs in boreal winter is the orographic GWD (OGWD) component. Directly above the subtropical jet, the OGWD component locally even constitutes the generally dominating zonal mean forcing (Šácha et al., 2019; Kuchar et al., 2020). Beyond this zonal mean perspective, OGWD is distributed in spatially asymmetric hotspots (Šácha

et al., 2018) with pronounced long and short term variability and intermittent strength (Kuchar et al., 2020). All of those aspects have implications for OGWD effects on dynamics and the methodologies for their analysis. Using a mechanistic model and focusing on a steady state response to localized OGWD perturbations, Šácha et al. (2016) demonstrated that the dynamical impact depends on the spatial distribution of the perturbation. Samtleben et al. (2019, 2020) clarified that the dynamical



response is sensitive especially to the distribution of OGWD relative to the phases of stationary planetary waves (PWs) that propagate from the troposphere to the stratosphere.

Using a set of dedicated CCM simulations, Kruse (2020) was able to track the evolution of the long term response to the OGWD, which results in polar cap potential vorticity anomalies. Kuchar et al. (2020) proposed a methodology based on composite analysis allowing to study a short-term response to the local intermittent OGWD peaks in a CCM. Using this methodology, Sacha et al. (2021) demonstrated that on the order of days, local OGWD peaks in the lower stratosphere influence the propagation of PWs from the troposphere to the stratosphere, either enhancing or suppressing the propagation towards the polar vortex depending on the hotspot. Especially the hotspot above the Himalayas has been discussed to show pronounced links to the polar vortex weakening. The study also explains, how lower tropospheric conditions immediately influence stratospheric dynamics via parameterized OGWD. Knowledge of how a particular GW hospot may influence the polar vortex morphology brings more confidence in predictability of SSW-like (weak polar vortex) events and potentially enhanced sub-seasonal predictability following weak polar vortex states (Baldwin et al., 2021), which is highly demanded, e.g. in the energy trading sector (Beerli and Grams, 2019).

In the present study we use the same CCM simulation and methodology as in Sacha et al. (2021) for a detailed analysis of polar vortex stability in connection with strong OGWD events in the Himalayas. We start with a model description and a brief description of observational datasets used in Sect. 2.1. In Sect. 2.2, we describe the construction of the Himalayas composite and in Sect. 2.3 all diagnostics that are used here. The results regarding the Himalayas impact on the polar vortex, ozone, and wave propagation are presented in Sect. 3. In Sects. 4, 5 and 6 we contextualise and discuss our results and provide concluding remarks.

## 2 Data and methodology

### 2.1 Description of model and observations

Following Kuchar et al. (2020) and Sacha et al. (2021), the present study is based on data from a specified dynamics simulation for the 1979–2010 period with the Canadian Middle Atmosphere Model (CMAM; McLandress et al., 2013). CMAM is a CCM with 71 levels in the vertical, which extends up to $7 \cdot 10^{-4} \, \mathrm{hPa}$ (about $100 \, \mathrm{km}$) with variable vertical resolution. It uses a triangular spectral truncation of T47, corresponding to a horizontal resolution of $2.5° \mathrm{x} 2.5°$, with the physical parameterizations being performed on a $3.75°$ horizontal grid. On spatial scales corresponding to zonal wavenumbers lower than 21, Newtonian relaxation ("nudging") is applied towards the 6-hourly horizontal wind and temperature field from ERA-Interim (Dee et al., 2011) up to $1 \, \mathrm{hPa}$ (see McLandress et al., 2014, for technical details).

In the present study, we evaluate the drag associated with freely propagating OGWs. OGWD in the parameterization is computed as a drag associated with two freely propagating hydrostatic zero-phase-speed GWs in the absence of rotation sourced by the subgrid scale orography respecting its anisotropy and flow direction. Moreover, low-level breaking induced drag and low-level drag associated with upstream blocking and lee-vortex dynamics are also parameterized within this three



component scheme (Scinocca and McFarlane, 2000). For information on the setting of tunable parameters of the scheme in the model we refer to McLandress et al. (2013).

A CMAM30-sd (specified dynamics) simulation was chosen for our analysis due to the evaluation and analysis of the strong OGWD composites by Kuchar et al. (2020). Providing freely accessible 6-hourly model data including 3D GW diagnostics,

CMAM30-sd represents a unique publicly available model dataset to investigate zonally asymmetric and interanually-variable wave torques. Moreover, CMAM is widely known for its realistic representation of middle atmospheric dynamics and has extensively been evaluated (Shepherd et al., 2014). In Sect. 3.3, we will contribute to this evaluation by assessing chemical and dynamical processes during strong OGWD events above the Himalayas in the CMAM30-sd simulation.

To verify the simulated Himalayas (HI) composite with observation-based datasets, we compare total column ozone (TCO)

of the CMAM30-sd simulation with the most recent generation of NASA's MERRA2 (Modern Era Reanalysis for Research and Applications-2, Gelaro et al., 2017) and ECMWF's ERA5 (Hersbach et al., 2020) reanalysis data. This comparison comprises a new way for future analyses using GW-resolving reanalysis such as ERA5 data, high resolution model simulations or long-term satellite observations.

## 2.2  Himalayas composite

Our study is based on strong OGWD event composites in the lower stratosphere for the GW hotspot above the HI mountain range, which is one of the dominant GW hotspots in the Northern Hemisphere (see, e.g., Kuchar et al., 2020). The HI hotspot refers to the region 70–102.5°E and 20–40°N at $70\,\mathrm{hPa}$ where the parameterized OGWD data was compressed to a single time series using spatial and daily averaging, respectively. The composite is computed using the peak-detection algorithm described in Kuchar et al. (2020). This algorithm detects local maxima of negative OGWD, which surpasses immediate neighbours

separated by more than 20 days with amplitudes beyond a normalized threshold, i.e. $-6.66\,\mathrm{m/s/day}$. The 20-day timescale complies with the definition of the World Meteorological Organization (WMO) criteria for SSW detection proposed by Charlton and Polvani (2007). The interpeak time distribution (see the interactive Html file in the supplement) reveals that the 20-day constraint lies close to the median time scale, and at the same time is a good representative of intra-seasonal timescales (up to $\sim 100\,\mathrm{days}$). The minimum of the interpeak time gap distribution is close to the 10-day timescale investigated in Kruse (2020).

While using the shorter timescale resulted in more detected events, the results are qualitatively similar.

The 10-day and 20-day timescales represent a way how longer timescales of the stratospheric variability are modulated by OGWs propagating from the troposphere with shorter timescales. The 20-day period is consistent with the internal mode resonance studied by Smith (1989), where the resonance of an atmospheric wave refers to wave enhancement due to positive reinforcement of a propagating wave in the stratosphere. Similarly, Sjoberg and Birner (2012) found that SSWs are preferentially

generated by a wave forcing on the 10-day timescale.

The statistical significance estimates and corresponding p-values of the composites were derived through application of a bootstrap method based on 10 000 samples. We use Jackknife resampling to estimate the probability uncertainty when each composite event was systematically left out and the probability calculated.



We evaluate significance fields using the minimum local p-values with global test statistics using the False Detection Rate
(FDR) methodology (Wilks, 2006), first described by Benjamini and Hochberg (1995) and later promoted by Wilks (2016) in
the atmospheric sciences. In addition to the hatching approach showing local p-values $< 0.05$ and $< 0.01$, respectively, we plot
boundaries of p-values $< 0.05$ corrected for FDR.

### 2.3 Diagnostics

To assess the impact of the HI hotspot on the polar vortex, we use the moment diagnostics defined by Seviour et al. (2013)
with the distinction that the vortex is identified using geopotential height on isobaric levels instead of potential vorticity on
an isentropic surface at $10\,\mathrm{hPa}$. We analyse three such moment diagnostics: aspect ratio, vortex centroid, and kurtosis. These
metrics indicate whether the vortex is stretched, displaced, or filamented, respectively. As we aim at identifying extreme vortex
events, the moment diagnostic thresholds were chosen according to Table 2 in Mitchell et al. (2011), i.e. 2.3, $72°\mathrm{N}$ and 1.7,
respectively.

Additionally, we compute the Northern Annular Mode (NAM) index at each altitude to differentiate between weak and
strong polar vortex events. The NAM at each level is defined as the first EOF of the zonally averaged zonal wind at $50-60°\mathrm{N}$
as adapted from Gerber et al. (2008). We set the NAM threshold to $-1$ to consider weak polar vortex events. Our NAM
definition explains 68.5% of the total variance at $10\,\mathrm{hPa}$ with a maximum variance of about 73.6% at $40\,\mathrm{hPa}$ (see Fig. S1 in
the Supplement).

Moreover, we use $u_{\mathrm{ratio}} \equiv \bar{u}/u_{\mathrm{REF}}$ at $60°\mathrm{N}$ and $10\,\mathrm{hPa}$ following Nakamura et al. (2020), who have put forward $u_{\mathrm{ratio}}$
as a local, instantaneous measure of the proximity to vortex breakdown (i.e., preconditioning). Here, $u_{\mathrm{REF}}$ is a wave-free,
reference-state wind inverted from the zonalized quasigeostrophic potential vorticity. For winters with SSWs $u_{\mathrm{ratio}}$ reaches
0.3, therefore we take a slightly larger value $u_{\mathrm{ratio}} = 0.4$ as a threshold for the non-SSW weak vortex events.

The response of resolved waves to OGWs is investigated using the Eliassen-Palm flux (EPF) diagnostics (Andrews and
McIntyre, 1987). The EPF convergence serves as an indicator of wave dissipation and the EPF divergence (EPFD) indicates
sourcing.

The propagation of linear RWs can also be diagnosed by means of the refractive index as, for example, in Simpson et al.
(2009), Eichinger et al. (2020), and first suggested by Matsuno (1970). RWs tend to propagate towards regions of large $n_k^2(\varphi, p)$
and cannot propagate through regions of negative $n_k^2(\varphi, p)$, which is defined as

$$n_k^2(\varphi, p) = \left[ \frac{\bar{q}_\varphi}{a(\bar{u} - c)} - \left( \frac{k}{a\cos\varphi} \right)^2 - \left( \frac{f}{2NH} \right)^2 \right] a^2. \tag{1}$$

Here, the overbars represent zonal means and the subscripts $_p$ and $_\varphi$ represent partial derivatives. $c$ is the zonal phase speed,
$N$ is the buoyancy frequency and $H = 7000\,\mathrm{m}$ is the density scale height. We assume $c = 0$ for quasi-stationary waves. The
meridional gradient of the potential vorticity $\bar{q}_\varphi$ is given by

$$\bar{q}_\varphi(\varphi, p) = 2\Omega\cos\varphi - \left[ \frac{(\bar{u}\cos\varphi)_\varphi}{a\cos\varphi} \right]_\varphi + \frac{af^2}{R_d} \left( \frac{p\theta}{T} \frac{\bar{u}_p}{\bar{\theta}_p} \right)_p, \tag{2}$$



where $f = 2\Omega\sin\varphi$ is the Coriolis parameter. We assume $k = 1$ to be the dominant zonal wavenumber in the stratosphere. $a = 6376 \cdot 10^3\,\mathrm{m}$ is the Earth radius, $R_d = 287.04\,\mathrm{J\,kg^{-1}\,K^{-1}}$, and $\Omega = 7.292 \cdot 10^{-5}\,\mathrm{s^{-1}}$. In our analysis, we present only the refractive index between 35–45°N and 200–70 hPa as in Wu and Reichler (2020), computed with a trimmed mean that excludes the top and bottom 10% of the values similarly to Watson and Gray (2015) and Weinberger et al. (2021). This representation of the refractive index located in the lower extratropical stratosphere is important for RWs propagating upward from the troposphere and actually overlaps with the region where the OGW drag dominates in the lower stratosphere.

In addition to the Eulerian diagnostics above, which are based on pressure coordinates, we calculate the "effective diffusivity" after Nakamura (1996) from the potential vorticity $q$ as a conservative tracer. The results are presented as the normalized equivalent squared which is a non-dimensional logarithmic quantity that provides the same information as $\kappa_{\mathrm{eff}}$ (Abalos et al., 2016). The effective diffusivity tends to be relatively large in mixing regions and relatively small in barrier regions (Haynes
and Shuckburgh, 2000). Note that barrier regions here act as eddy transport barriers and not as barriers for the advective circulation (McIntyre et al., 1995).

## 3 Results

### 3.1 Impact on the polar vortex

Fig. 1 presents the lag analysis of the polar vortex morphology for the HI composite. Four different metrics are studied, namely
the aspect ratio (**A**), centroid latitude (**B**), kurtosis (**C**), and proximity to vortex breakdown $u_{\mathrm{ratio}}$ (**D**).

From Fig. 8 in Kuchar et al. (2020), we know that for strong OGWD events in the Himalayas, OGWD anomalies are pronounced four days before and after the peak OGWD, which is due to the high intermittency of OGWs. The aspect ratio (see Fig. 1**A**) starts to increase six days before the peak, when the HI hotspot is already active. The composite average does not reach the threshold for a substantially stretched vortex (aspect ratio = 2.3), however, the shaded 95% confidence interval shows
that it can happen between day 3 and 4 after the OGWD peak. This suggests that the strong OGWD events in the HI hotspot can have an effect on the geometric shape of the vortex.

Sacha et al. (2021) argue that the dominant OGWD effect is indirect, acting via the modification of resolved wave propagation, in particular the leading wave modes.. It is reasonable to expect such an indirect effect to manifest with some time delay, presumably of a few days, which is about the time scale of the RW breaking (Dickinson, 1970; Cohen et al., 2014).
Fig. 1**B** shows an equatorward shift of the centroid latitude, indicating a vortex displacement. Before the strong OGWD event, the centroid latitude is shifting slightly northward, during the event the latitude stays approximately constant, and around 4 days after the event, a pronounced equatorward shift emerges. At lag=9, the vortex displacement can reach to latitudes as low as 72°N.

In addition to the vortex being stretched and/or displaced, the kurtosis (see Fig. 1**C**) can indicate the types of specific polar
vortex events. Negative values of the kurtosis indicate a vortex splitting event, while high positive kurtosis values indicate high filamentation, which may accompany both splitting and displacement events (Mitchell et al., 2011). In the light of this,



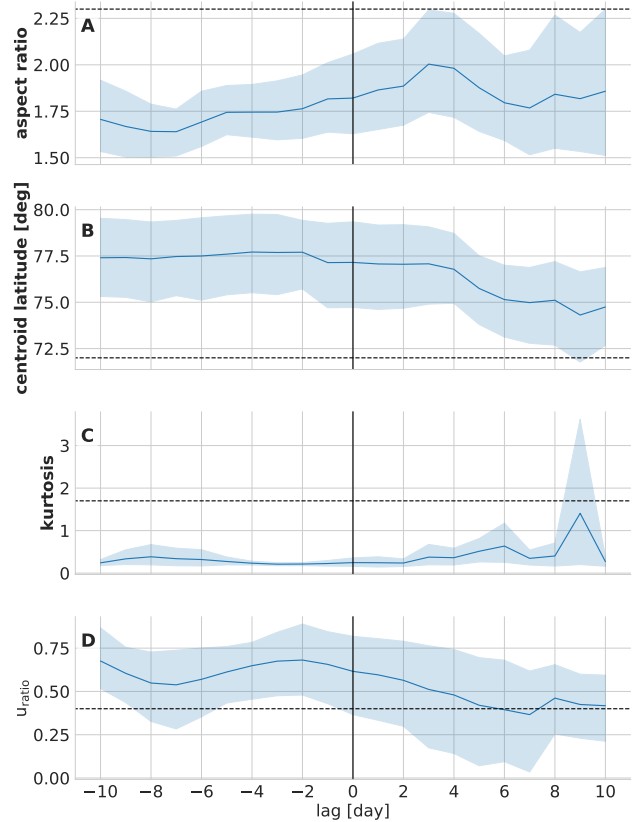

**Figure 1.** Composites documenting vortex morphology (aspect ratio (**A**), centroid latitude (in degrees; **B**), kurtosis (**C**), proximity to vortex breakdown (**D**)) in absolute values. Black dashed lines illustrate particular thresholds: 2.3, 72°N, 1.7, and 0.4, respectively. Shading represents 95% confidence intervals calculated via bootstrapping.

the kurtosis composite follows the time evolution of the aspect ratio and centroid latitude, with a sharp positive peak at lag=9 indicating eventually a polar vortex split event as further documented in Sect. 3.

The proximity of a vortex breakdown is depicted by $u_{\mathrm{ratio}}$ in Fig. 1**D**. The ratio decreases after the start of the strong OGWD 185 event and a part of the shaded region crosses the threshold at lag=0. The decrease continues for positive lags and the mean value of the ratio reaches the threshold at lag=6 and a minimum at lag=7. Afterwards the mean value slightly increases, however, at the same time the composite variability strongly decreases and the ratio values of shaded regions continue to decline. Overall, the variability of this quantity is relatively large (broad shaded region), and for the lower bound of our composite mean estimate the vortex is close to breakdown from lag=0 onwards, peaking at lag=7. At lag=9, when we have indications of a polar vortex 190 disruption from the other metrics, the mean ratio stays slightly above the threshold, but the composite variability is reduced. The composite mean estimate being close to the threshold indicates that the vortex may be robustly preconditioned for a breakdown.





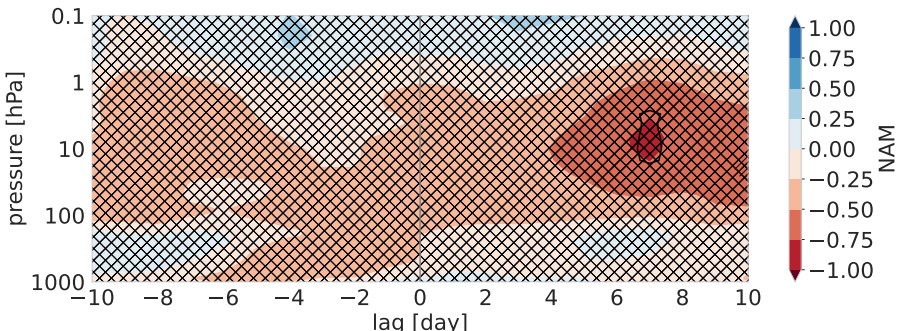

**Figure 2.** Composite anomalies of NAM. Red and blue colors represent weak and strong polar-vortex events, respectively. Hatching \\\\ and //// represent p-values $< 0.05$ and $< 0.01$, respectively. Black contour lines align p-values $< 0.05$ obtained after FDR.

Figure 2 shows the composite NAM anomalies as a function of pressure. Here, a strong OGWD event does not have any significant immediate effect on the NAM. From lag=4 onwards, an indirect response has built up, manifesting in a significant
negative NAM anomaly in the middle stratosphere. The minimum of the NAM anomaly appears around lag=7, in correspondance with the $u_{\mathrm{ratio}}$ minimum, suggesting a weakening of the polar vortex. The significant NAM anomalies (hatched), is confirmed by the FDR methodology (black contour). There is no statistically significant NAM signal in the troposphere, suggesting no influence on stratosphere-troposphere coupling by the strong events in the Himalayas.

A probabilistic analysis can provide an additional tool for describing the variability of the composites. Fig. 3 shows the
probability of the occurrence of weak polar vortex situations around the strong OGWD events above the Himalayas. These events are defined as NAM values less or equal to -1. Shortly before the start of a strong OGWD event, the probability is low. It further drops, when the event initiates, but from lag=-1 a monotonic increase of the probability starts, which maximizes between 5 and 10 days after the OGWD peak. This means that at that time, it reaches up to 50 % probability of the weak vortex, likely due to the combination of both direct and indirect OGWD effects. This corresponds with the significant NAM anomaly
found in Fig. 2.

Overall, the results show that the strong OGWD events above the Himalayas are not an ultimate prerequisite for the vortex breakdown. However, all the indicators demonstrate that they help to precondition the polar vortex with a preference towards the split event as suggested by the moment diagnostic composites. Using longer datasets, evaluation of the probabilistic perspective of a weak polar vortex when filtered according to the phase of Quasi-Biennial Oscillation (QBO) may show even more
promising results (see Fig. S4 in Sacha et al., 2021).

## 3.2   Impact on resolved waves

In this section, we analyze dynamical mechanisms related with the polar vortex weakening discussed in Sect. 3.1, in particular, how the resolved wave forcing responds to OGWD strong events. Fig. 4 shows the composite of EPFD anomalies after the strong OGWD event.





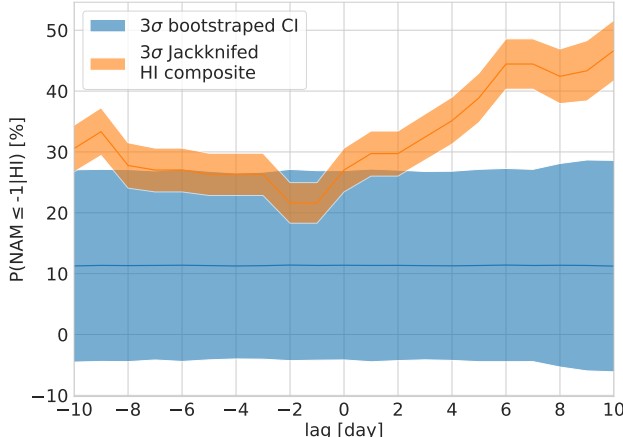

**Figure 3.** Conditional probabilities that NAM is less or equal $-1$ given strong OGW events above the Himalayas. Blue shading represents $\pm 3\sigma$ confidence intervals calculated via bootstrapping. Orange shading illustrates probability uncertainty calculated via Jackknife resampling.

Negative EPFD at different altitudes anomalies can be seen in the stratospheric high latitudes at lag=0, 3 and 5. The negative EPFD anomaly at lag=0 in the polar stratosphere for the HI composite (see Fig. 4**A**) was already highlighted in Sacha et al. (2021) (see their Fig. 1a) and was attributed to an OGWD effect resulting in enhanced PW (with zonal wavenumber 2) propagation (see also Fig. 4**I**). At 30 hPa and lag=5 (see Figs. 4**G** and **K**) we observe negative EPFD anomalies for the wavenumbers 1 and 2 positively interfering in a statistically significant anomaly in EPFD for all waves. This significant anomaly precedes the minimum of the NAM around lag=7.

The positive EPFD anomaly south of the hotspot at 100 hPa indicates an increased sourcing of PWs with zonal wavenumber 2 above the tropopause. Additionally, we observe a positive EPFD anomaly above the hotspot at 10 hPa for PWs with zonal wavenumber 1. This documents that not only propagation but also sourcing of planetary-scale waves may be modulated by OGWs. PWs can be spontaneously generated by baroclinic instability (Hartmann, 1979) which has been shown to be contributed by GWs (Sato and Nomoto, 2015). Another mechanism how to generate PWs in situ has been demonstrated for the upper stratosphere and mesosphere due to zonally asymmetric GW forcing (Holton, 1984; McLandress and McFarlane, 1993; Song et al., 2020). As already shown in Sacha et al. (2021), the estimated nudging strength is negligible at the analysed levels compared to the resolved wave forcing, hence the anomalies are a result of internal model dynamics.

## 3.3 Impact on ozone

In the sections above, we have shown that strong OGWD events over the HI hotspot weaken the polar vortex and alter the PW field. Sacha et al. (2021) argued that the dynamical response to the strong OGWD events is highly transient, and hence does not allow for a direct application of the traditional residual mean framework to assess the effects on zonal mean transport. In Fig. 5 we show the local evolution of the transient Rossby wave response by means of the finite-amplitude Local Wave



**Figure 4.** Composite anomalies of the Eliassen-Palm flux $\boldsymbol{F}$ (arrows; units: $(1;10^{-2})\,\mathrm{kg\,s^{-1}\,s^{-1}}$) and its divergence $\nabla \cdot \boldsymbol{F}$ (shading; units: $\mathrm{m\,s^{-1}\,day^{-1}}$) and zonal mean OGWD anomalies (gray solid (positive) and dashed (negative) contours: $\pm 0.1, \pm 0.5, \pm 1, \pm 3\,\mathrm{m\,s^{-1}\,day^{-1}}$, linewidths increase with increasing values) at lags 0, 3, 5, 7 days for the Himalaya. In panels E–P the anomalies are decomposed into leading zonal planetary wave modes. The green horizontal and vertical lines represent regions of the particular GW hotspot and the $70\,\mathrm{hPa}$ level. The black dashed line denotes tropopause pressure of the respective composite. The EPFD anomalies are colored only where the p-values of the anomalies are $< 0.05$. Black contour lines aligns p-values obtained after FDR.





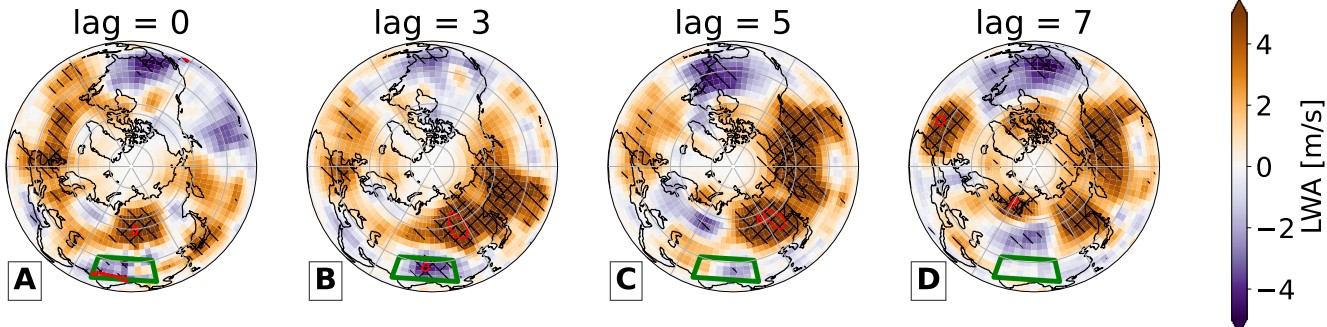

**Figure 5.** Composites of finite-amplitude Local Wave Activity anomalies [m/s] in CMAM30-sd at lags 0 (**A**), 3 (**B**), 5 (**C**) and 7 (**D**) days, respectively. The green box represents the Himalayas (70-102.5°E and 20-40°N) hotspot. Hatching \\\\ and //// represents p-values $< 0.05$ and $< 0.01$, respectively. Red contour lines aligns p-values obtained after FDR.

Activity (LWA) composites at different lags (Huang and Nakamura, 2016). LWA can be broadly viewed as a measure of the horizontal flow meandering, however large LWA values can also mark regions of major Rossby wave breaking. At lag=0, pronounced and significant anomalies are localized around the HI hotspot, with negative wave activity anomaly in the region of the OGWD and positive anomaly at the northern flank of the hotspot, probably connected with over and around flowing of the OGWD region. At lag=3 the magnitude of both anomalies is larger with the positive LWA anomaly being advected and spreaded further eastward and northward. By lag=5 the positive anomaly reaches the western coast of North America and by lag=7 it also spreads over the polar region.

These LWA anomalies turn out to be perfectly correlated with the distribution of TCO composite anomalies in Fig. 6. It shows a similar pattern of a negative anomaly inside the hotspot and a pronounced positive TCO anomaly at the northern flank at lag=0 reaching into the polar vortex region. TCO shows values above 400 DU northeast of the HI hotspot and these values as well as the pattern itself agree well with the TCO composites of MERRA2 and ERA5 (Figs. S2**B** and **C**, respectively). The positive anomaly north of the hotspot, is also moving eastward with time, and as the lag increases, we see a buildup of the positive anomaly in the polar region that evolves to the maximum strength at lag=7. However, we must note that the LWA anomalies were shown at 70 hPa, but only the TCO anomaly north of the hotspot is connected with anomalous ozone concentrations in the lower stratosphere. The TCO anomaly in polar region is connected with anomalous ozone concentrations in the upper stratosphere and lower mesosphere (see Fig. S3). The correspondence of LWA anomalies with local TCO anomalies around the hotspot can be explained by the quasi-horizontal flow (and consequently ozone concentration contours) meandering in the lower stratosphere. However, the co-occurence of LWA and TCO anomalies in the polar vortex region may suggest a possible Rossby wave breaking and induced mixing.

To test this hypothesis, we show in Fig. 7 the effective diffusivity $\kappa_{\text{eff}}(\phi_e)$ at 450 K ($\sim$18 km). This diagnostic can help to identify transport barriers and regions of enhanced mixing (see e.g. Abalos et al., 2016; Eichinger et al., 2019). Indeed, Fig. 7 shows enhanced diffusivity around $\phi_e = 60°$N, i.e. around the polar vortex boundary (depicted by the thick line in Fig. 7).



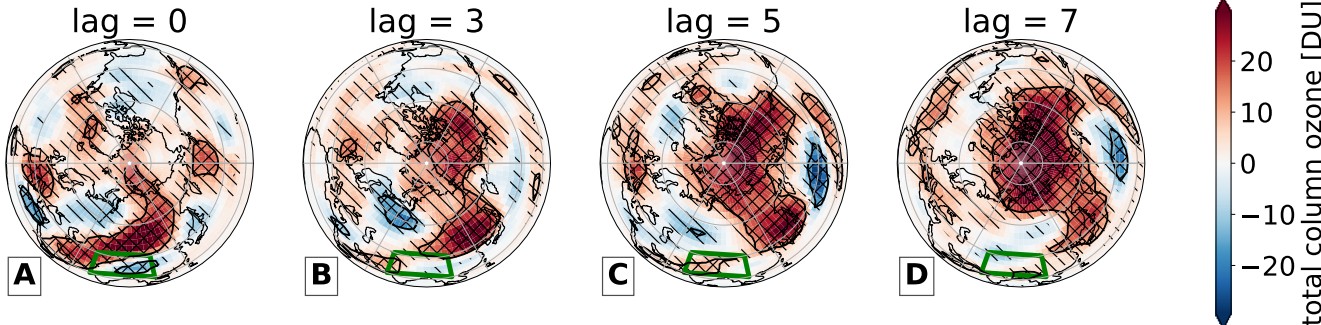

**Figure 6.** Composites of anomalies documenting evolution total column ozone [DU] in CMAM30-sd at lags 0 (**A**), 3 (**B**), 5 (**C**) and 7 (**D**) days, respectively. The green box represents the Himalayas (70-102.5°E and 20-40°N) hotspot. Hatching \\\\ and //// represents p-values $< 0.05$ and $< 0.01$, respectively. Black contour lines aligns p-values obtained after FDR.

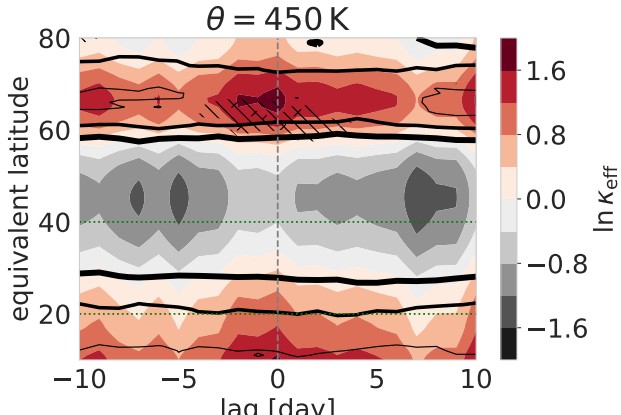

**Figure 7.** The logarithmic distribution of $\kappa_{\mathrm{eff}}(\phi_{\mathrm{e}})$ composite anomalies at 450 K as a function of equivalent latitude $\phi_{\mathrm{e}}$ and lag [days]. Black contours increases their linewidths with values of the climatological composite of 1, 1.5, and 2, respectively. These contours differentiate between regions of low, moderate, and strong mixing, respectively. Hatching \\\\ and //// represents p-values $< 0.05$ and $< 0.01$, respectively. The gray dashed line illustrates lag=0 when the OGWD composite peaks. Green horizontal lines illustrate latitudinal bounds of the Himalayas.

These enhanced mixing values start already with the start of the OGWD event at lag=-3 and have a peak around lag=0. The statistically significant anomalies then last up to lag=5. This supports the hypothesis that strong OGWD events influence the TCO anomalies in the polar vortex due to the anomalous lower stratospheric mixing into the polar vortex region. After the quasi horizontal mixing into the polar vortex, the anomalous ozone is redistributed by transport processes inside the vortex. de la Cámara et al. (2018) have shown that mixing increases similarly at and after the central date of SSWs around $\phi_{\mathrm{e}} = 60°\mathrm{N}$. Note that after FDR there are no p-values $< 0.05$ in contrast to LWA and ozone fields in Figs. 5 and 6, respectively.



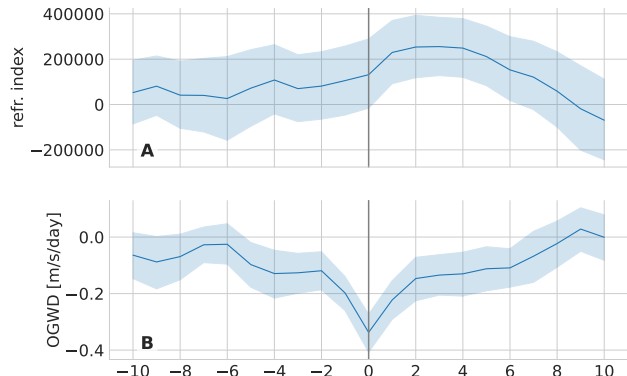

**Figure 8.** Composite anomalies of refractive index between 35–45°N and 200–70 hPa (**A**) and zonally averaged OGWD $[\mathrm{m\,s^{-1}\,day^{-1}}]$ between 20-40°N at 70 hPa (**B**). Shading represents 95% confidence intervals calculated via bootstrapping.

## 4  Contextualization

Model biases in SSW frequency are strongly controlled by PW propagation. Wu and Reichler (2020) showed that in CMIP5 and CMIP6 models the index of refraction for PWs in the lower stratosphere just above the subtropical jet is a key factor for
controlling PW propagation to the stratosphere. This is a region of large model uncertainties (see Fig. 6 in Wu and Reichler, 2020) and it overlaps with the lower stratospheric area where parameterized OGWD dominates (see Fig. 1 in Kuchar et al., 2020) and where we here analyze the strong HI OGWD composites. Fig. 8**A** shows the refractive index composite time series around the strong OGW event. A strong increase of the refractive index can be seen in the course of the strong OGWD event with a maximum between lag=2 and 4. This corresponds with the EPFD maximum at 100 hPa (see Fig. 4**C**).

The refractive index is positively correlated with the frequency of SSWs in the CMIP models, with a local minimum of the refractive index acting as a valve controlling the vertical wave propagation into the stratosphere. This is exactly the leading effect of parameterized OGWD influence on resolved wave propagation (Šácha et al., 2016; Eichinger et al., 2020; Sacha et al., 2021). Recently, the link between stronger OGWD, weaker EPFD and a too strong polar vortex has been demonstrated by Hall et al. (2021) for CanESM5.

To put the strong localized OGWD events above the Himalayas in the zonal mean forcing perspective, Fig. 8**B** shows the composite anomaly time series of zonally averaged OGWD, demonstrating how locally strong OGWD events above the Himalayas manifests in the zonal average. The zonally averaged OGWD between 20 and 40°N at 70 hPa drops around five days before and rises about five days after the onset. In comparison with Fig. 5 in Kuchar et al. (2020), it constitutes about 30% of the climatology of zonally averaged OGWD in CMAM30-sd. Owing to the DynVarMIP iniative (Gerber and Manzini, 2016) at
least zonally averaged GW characteristics at daily resolution are available from CMIP6 model simulations (three dimensional GWD as a monthly mean only). For future work we plan to apply our peak detection method in the context of uncertainty in polar vortex projections (Manzini et al., 2014; Ayarzagüena et al., 2020).



## 5   Discussion

The separation between the strong OGW events vary from the 10-day to the 100-day timescale, i.e. on the intra-seasonal
timescales. The median time scale of 20 days used for the HI composite is consistent with the internal stratospheric resonance
mode (Smith, 1989) responsible for two thirds of SSWs (Birner and Albers, 2017). Overall, the variability of OGWs on
such short timescales actually represent a way how longer timescales of the stratospheric variability are modulated by OGWs
propagating from the troposphere.

Vortex splitting events are generally associated with anomalously high wavenumber-2 PW activity in the stratosphere (Matthew-
man et al., 2009). Similarly to Mitchell et al. (2011) we observe that the stretching tendency is accompanied with the centroid-
latitude tendency to equatorward values and the kurtosis may become largely positive as well. This aligns with findings of
Albers and Birner (2014) that during split events planetary and GWs tune the vortex geometry toward its resonance.

The increasing evidence for a link between polar vortex dynamics and parameterized OGWD (with all its simplifications)
stimulates an exciting future research question, namely whether there is a relationship between (O)GWD uncertainty in the
stratosphere and the frequency of SSWs. If that should be the case, it would be essential to capture the small-scale perturbations
by GWs in current generation Earth System Models for predictions and climate projections of the polar vortex and for SSWs.
Li et al. (2008) hypothesized that zonally averaged OGWD in the lower stratosphere may be modified because of climate
change-induced trends in wind and the static stability in the upper troposphere/lower stratosphere region. The analyzed nudged
CMAM30-sd simulation does not show any significant OGWD trends, because its magnitude and distribution show high
interannual variability (Šácha et al., 2018). In the CMAM CCMI-REFC1 simulation, however, a negative trend in the valve
region was found, even after correction for the upward shift of the tropospheric circulation (Šácha et al., 2019). Given also the
sensitivity of the OGWD effect on the background conditions, this may be one of the factors behind the uncertainty in SSW
projections in future climates (Ayarzagüena et al., 2020).

While our composite shows promising results in terms of the NAM probability, as a standalone metric it would suffer from
numerous false alarms. Using longer datasets, potentially with resolved GWs, it would be desirable to evaluate the HI composite
in combination with other precursors such as QBO or Madden-Julian Oscillation (Domeisen et al., 2020). Our results rather
aim at highlighting an area where model improvement would be beneficial to overall predictability bit by bit. In future work a
standalone metric could be developed with even lower false alarm rate.

Dietmüller et al. (2021) discussed in detail the model spread in ozone variability and trends in the lower stratospheric mid
latitudes, however, they could not identify a clear cause for this spread. Our results now show that also the representation
of GWs have the potential to alter ozone variability and trends in the lower stratospheric mid latitudes in chemistry climate
models and hence can play a role for the inter-model spread. In further consequence, enhanced intrusion of ozone into higher
latitudes result in intensified cooling due to longwave radiation during boreal winter but the adiabatic heating due to the
enhanced residual circulation plays a key role in lower stratospheric temperature anomalies. Moreover, the GW-induced ozone
anomalies can influence the impact of stratospheric intrusions on surface ozone, in particular during winter and spring (Monks,
2000; Knowland et al., 2017).



Recent studies show that ozone can be influenced by non-dissipative effects of GWs (e.g. Chang et al., 2020; Gabriel, 2022). However, as these processes are not resolved in the here used model simulation and neither are included in the parameterization, this possible influence cannot be analysed here. This motivates to further study the influence of GWs on trace gas distributions, in particular with respect to ozone. For this, high resolution reanalysis data such as ERA5 and GW-resolving model simulations could be used.

## 6 Conclusions

In this study, we examine implications of how strong OGW breaking above the Himalayas in the lower stratosphere impact the polar vortex morphology. To composite the strong OGWD events, we apply a recently developed method detecting these strong OGWD events (Kuchar et al., 2020; Sacha et al., 2021) parameterized in a specified dynamics simulation of the chemistry-climate model CMAM. The impact on the polar vortex is described via classical zonally averaged diagnostics such as the EP flux framework in addition to the moment diagnostics taking into account the polar vortex spatial morphology. These moment diagnostics indicate whether the vortex is stretched, displaced or filamented, respectively. However, we include also novel frameworks to diagnose the vortex preconditioning (Nakamura et al., 2020).

We show that an imposed drag in the lower stratosphere by the Himalaya GW hotspot (although much weaker in magnitude than in the mesosphere) located at the edge of the surf zone can profoundly influence the polar vortex. Using the lead-lag analysis, we show that strong OGW drag events above the Himalayas are associated with anomalously increased upward RW propagation and RW breaking in the stratosphere. The accompanying polar vortex stretching results from the enhanced PW with zonal wavenumber 2 discussed in Sacha et al. (2021). While the diagnostics may not meet the thresholds previously defined for the the vortex breakdown, we demonstrate that these strong OGWD events above the Himalayas help to precondition the polar vortex and increase the probability of the weak vortex state up to 50%.

The association between SSWs and strong OGWs breaking above the Himalayas is put into context via the refractive index in the lower stratosphere shown as important factor for the SSW frequency in the CMIP models (Wu and Reichler, 2020). Our results also prove the potential of GWs to alter the ozone variability and its trends via mixing in the surf zone or advection from lower latitudes. The evidence for a link between polar vortex dynamics and the parameterized OGWD via PWs stipulates further research of the uncertainty in polar vortex projections (Manzini et al., 2014; Ayarzagüena et al., 2020) or ozone variability (Dietmüller et al., 2021).

Given all the assumptions and tuning employed in the OGW parameterization scheme, this indication of surprisingly wide ranging effects of parameterized OGWD in the model prompts validation in GW resolving datasets and makes an additional case for improving of the parameterizations in CCMs.



*Code and data availability.* All processed data files for this study are provided via Mendeley Data (Kuchar, 2022b) in addition to previous publications (Kuchar, 2020b, c). All codes to reproduce our figures are provided via GitHub (Kuchar, 2022a) in addition to previous publications (Kuchar, 2021, 2020a).

*Author contributions.* AK, PS and RE designed the study. AK analysed the data. AK, PS and RE compiled the manuscript with inputs of all
other authors.

*Competing interests.* The authors declare that they have no conflict of interest.

*Acknowledgements.* AK and CJ acknowledge support from Deutsche Forschungsgemeinschaft under grant JA836/43-1. PP and RE are supported by GA CR under Grant Nos. 21-20293J and 21-03295S. The authors would like to thank all colleagues involved in the CMAM30-sd model simulation (obtained from http://climate-modelling.canada.ca/climatemodeldata/cmam/output/CMAM/CMAM30-SD/index.shtml,
last access: 25 September 2020).

Furthermore, we acknowledge developers of python open-source software libraries used for this paper: *aostools* (Jucker, 2018), *cartopy* (Met Office, 2010 - 2015), *hn2016_falwa* (Huang et al., 2022), *matplotlib* (Hunter, 2007), *numpy* (Oliphant, 2006), *pandas* (McKinney, 2010), *scipy* (Virtanen et al., 2020), *seaborn* (Waskom et al., 2016), *statsmodels* (Skipper Seabold and Josef Perktold, 2010), *xarray* (Hoyer and Hamman, 2017), and *xcontour* (Qian, 2022).





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
