# Peer review of "On the impact of Himalaya-induced gravity waves on the polar vortex, Rossby wave activity and ozone"

_EGUsphere, 2022_

## Author Comment (AC2)

We thank the reviewers for their comments, which helped us in improving the quality of the paper. We have revised the manuscript according to their hints. Below we repeat the reviewer´s points of concern with our respective responses below. We hope the paper is now suitable for publication in ACP.

**Referee #1**

**The methodology mainly consist in doing composite maps according to the orographic gravity wave tendency applied at 70hPa averaged over the Himalayas. Composite are made of the NAM, the EP fluxes, refractive indexes of Rossby wave activity. These are very specific diagnostics, picked up at convenience, to a certain extent, the authors trying to make a picture out of them.**

We cannot fully appreciate this comment, as the diagnostics used here, especially the Eliassen-Palm flux and refractive index are key quantities for examining the stratospheric dynamics and their importance for our understanding of the stratosphere has been demonstrated in numerous research articles as well as textbooks (e.g. Andrews et al. (1987)). The Northern Annular Mode (NAM) index is also a traditional quantity and is shown in the paper to supplement the vortex-oriented quantities with information on the impact on the mid-latitudinal stratosphere.

**As confidence intervals in the maps are often quite weak (95% significance occurring out of 20 days of realisations in Fig. 1A, can simply be due to chance) the quantitative arguments should be made stronger. In fact, it well may be that the large scale flow changes seen are manifestation of a strong mountain torque, due to large scale Rossby wave mountain interaction, and that the associated large scale conditions leads passively to a large subgrid scale Himalayan torque. These types of passive relation are not considered at all in the paper, and should be discussed.**

In the paper, we analyze the OGWD impact on the stratospheric vortex.

In the troposphere, the Rossby wave-mountain interaction is inherently present in our composites and it (among other processes) naturally impacts the amount of momentum flux propagated by freely propagating OGWs within the OGW parameterization scheme. Also, this situation in the troposphere is reflected in the variations of the Rossby wave propagation upwards in the troposphere. Šácha et al (2021) discussed the term hotspot preconditioning

which also involves stratospheric background processes that can cause a critical level for OGWs.

Additionally, we would like to acknowledge that the OGWD is the dominant force in the lower stratosphere in the so-called valve layer (Kruse et al, 2016). It controls the Rossby wave propagation from the troposphere to the stratosphere. This is documented in the literature and also demonstrated by our results specifically for the HI hotspot.

We now make clear in the paper (see l375-379) that the final impact on the stratosphere captured by our composite consists of the background tropospheric conditions that set the base wave flux from the troposphere and also nonlinearly influence the occurrence of the strong OGWD event, and from the direct OGWD impact, which is mainly pronounced as alternation of the Rossby wave propagation from the troposphere to the stratosphere.

As an illustration, the composite maps in Fig.4, and Fig.5 never shows the negative lags, it would be important to see if something is present at negative lag before discussing causality.

We included Figs. 4,5,6 with negative lags in the revised manuscript or its supplement and discuss them accordingly.

Also, there exist simulations CMIP type among others, where centuries of TEM data are provided, far much more than the 30 years shown here, the authors should consider these type of simulations to consolidate their correlations: 95% confidence out of 30 years low resolution runs are a little out of date according to the present day standards. Also looking at other models could tells if the OGW in CMAM are representative of what is parametrized in other models.

We agree with the reviewer that the statistical significance in a small sample size does not mean the results are robust enough. While CMAM30-SD may be considered an outdated dataset, currently there is no alternative dataset available with sufficient outputs (3D GWD with daily temporal resolution) to our knowledge, and also not considered in DynVarMIP (Gerber and Manzini, 2016). Therefore, we suggested (see l296-304) that zonally averaged OGWD may be analyzed to investigate the impact of large torques using long CMIP6 simulations.

Many caveats are placed in the conclusion, somehow in line with what is said above about correlations not necessarily implying causality, but the titles and abstract are not

that humble and strongly suggest that a particular model behaviour translates something that occurs in reality.

We agree with the reviewer, so we softened our findings in the conclusion, title and abstract and state that our findings are related to effects of parameterized OGWD in the CMAM30-SD simulation in particular and need to be verified in future studies.

**Referee #2**

This study investigates flow changes in the boreal winter stratosphere following parameterized orographic gravity wave drag (OGWD) events over the Himalayas at 70 hPa, using a 30-year simulation of a chemistry climate model where temperature and wind have been nudged to ERA-Interim fields. Based on composites of different values around OGWD events, it is claimed that these OGWD events alter the propagation of planetary waves in the lower stratosphere and the morphology of vortex, preconditioning the vortex for a subsequent breakdown. It is also claimed that the alteration in the planetary waves have impacts on planetary wave breaking and large-scale stirring in the lower stratosphere, affecting the concentration of high latitude ozone.

I find that the figures shown do not allow a proper analysis of the problem, as it is explained in my comments.

Major comments:

1) The methodology of compositing around OGWD events over the Himalayas does not necessarily imply that these events are the cause of the circulation alterations shown on a global scale. The argument given by the authors is that the potential connection between increased OGWD over the Himalayas and vortex alterations is mediated by changes in planetary wave propagation and dissipation. Figure 4 is supposed to prove this point, but there are several reasons why this is not achieved: 1) the EP flux vectors are not well scaled and they are dominated by the horizontal component, so it is not possible to analyze the anomalies properly. Please see Edmon et al (1980) and/or Jucker (2020) to scale the vectors correctly. 2) Only positive lags are shown. This is an important point, since it would be perfectly possible that the parameterized GW events are the result of large-scale changes of the background flow and/or planetary waves. So claiming that EP flux anomalies at lag 0 are the result of simultaneous OGWD events over the Himalayas, without any further analysis, is unconvincing.

We completely revised Fig. 4 so that the EP-flux vectors are scaled appropriately and negative lags are shown as well. We discussed the revised figure accordingly. Also, we point

this reviewer to our response to Ref#1, where we explain the causality between large-scale circulation in the troposphere, stratosphere and the parameterized OGWD.

**2) About the impact of OGWD events on vortex geometry and variability.**

Figure 1. The evolution of the vortex geometry parameters are not compared to their mean seasonal evolution, and hence we cannot tell if the evolution is statistically different from the mean seasonal evolution. Showing the 95% confidence intervals does not give any useful information in this respect unless the composites show deseasonalized anomalies instead of the global value.

The vortex geometry parameters are not shown as deseasonalized anomalies because we check whether they meet thresholds for extreme vortex events as in Table 2 in Mitchell et al (2011). Nevertheless, the bootstrap independence of the vortex geometry parameters with respect to the climatological evolution for p-values < 0.05 is visualized using dots. At lag=3 we can see that the aspect ratio is statistically significant from climatology, thus, indicating a polar-vortex stretching. At lag=9 centroid latitude and kurtosis are statistically significant from climatology. We include this figure and its description in the revised manuscript.

[Figure]

In lines 179-192 there are strong claims about the vortex being "robustly preconditioned for a breakdown", and "indicating a polar vortex split". This should be easy to quantify: How many of these parameterized OGWD events are followed by SSWs? How many SSWs are preceded by OGWD events? Also, the time evolutions shown in Figs 1-3 stop at lag +10 days; I strongly suggest to show more positive lags, so as to be able to analyze if significant alterations to the vortex indeed take place.

In CMAM30-sd we have detected 16 SSWs based on vortex moments (Seviour et al, 2013), respectively. When the begin date of an SSW event is preceded by a strong OGWD event, we have detected the following amount of SSWs:

within 30 days: 6

within 40 days: 8

within 60 days: 9

According to a binomial probability test, the null hypothesis that SSWs associated with strong OGWD events above the Himalayas happen by chance (i.e. with 50% probability) cannot be rejected at the 5% level of significance for all separation intervals because the returned p-value is greater than the critical value of 5%.

While we might have been too strong in our claims that the vortex is being "robustly" preconditioned for a breakdown, we have not mentioned SSWs intentionally and we rather associated the strong OGWD events with weaker vortex states through the NAM as Fig. 3 and support this claim with the probabilistic view in Fig. 4. We stress this statement in l134.

Now we show lags from -10 to 10. It corresponds to the 20-day constraint within the detection algorithm. The suggested extension beyond +10-day limit could possibly alias with other OGWD events due to their large intermittency and we are cautious to do so.

**3) About the impacts on ozone.**

I do not follow very well the arguments trying to explain the connection between the TCO anomalies and the wave activity anomalies at 70hPa and effective diffusivity at 450K, in turn being "caused" by the parameterized OGWD events. The first thing is that the spatial patterns of TCO and wave activity are said to be "perfectly correlated", and are clearly not. Aside from that, the argument is that strong finite-amplitude wave activity may be signaling planetary wave breaking, and this stirs and mixes trace gasses. And this is why effective diffusivity is shown, as a metric for irreversible mixing. However, significant anomalies of all these fields are shown at lag 0, so it is difficult to argue that at lag 0 the planetary waves have already had enough time to alter their propagation and breaking, and produce an impact on TCO. In lines 172-174, the authors already acknowledge that "it is reasonable to expect such an indirect effect to manifest with some time delay, presumably of a few days", when talking about the proposed mechanism for the OGWD impacting on the planetary waves.

We may have gotten a little overwhelmed by how well the patches agree, i.e. the quantities correlate (negative anomalies of TCO and LWA inside the hotspot and a pronounced positive

anomaly of TCO and LWA at the northern flank at lag=0), of course this is not perfect, we changed the statement to „well correlated" (l256).

About the time lag, please note that lag=0 is just the climax of the OGW peak events, significant anomalies already start 4-5 days before lag=0 (see Fig. 8 of Kuchar et al., 2020). This is hence just the right timing for Rossby wave propagation and breaking alterations to adjust and impact TCO. Furthermore, the transport associated with the mixing around lag=0 is rapid compared to the time taken for horizontal advection at later lags (Plumb, 2007; Butchart, 2014). Thank you for pointing this out, we forgot to mention this in the text, but now add an explanation (l272-277). Additionally, clarity and visibility of the issue should now be improved through the proposed figures showing negative lags as well.

Figure 7 - effective diffusivity at 450K. The isentropic level of 450K is located in the lowermost stratosphere, and I believe it is not very useful to understand variations in the TCO. At this level, there is more ozone concentration at high than at low latitudes, so increased ozone mixing would smooth the latitudinal gradients by reducing otherwise high concentrations over high latitudes, and increase otherwise low ozone concentrations over low latitudes. These effects on O3 mixing rations are captured in zonal mean ozone anomalies around 100-150 hPa (Fig. S3 in the supplementary material).

As we indicate in the text and show in Fig. S3, the TCO anomalies originate from various altitudes, partly depending on latitude. As discussed, mid-latitude TCO anomalies clearly come from the lowermost stratosphere (see former Fig. S3, current Fig. 4), it very well makes sense to use the 450 K isentropic level, because this level is located in the lowermost stratosphere.

The reviewer correctly points out that we have confused the direction in which mixing influences the above-mentioned TCO anomalies. Due to the ozone gradient, the mechanism must work the other way round. As isentropic mixing is a two-way process, this does not influence our general idea, it does, however, change the particular explanation. The ozone anomalies seem to evolve from enhanced out-mixing from the polar vortex, rather than from enhanced in-mixing into the vortex. This is additionally supported by the negative ozone anomaly in the high latitude lowermost stratosphere in Fig. S4. We changed this accordingly in the manuscript (l276) and we thank the reviewer for indicating this mistake.

4) The organization of the results is not intuitive: The first subsection presents the evolution of different metrics of vortex geometry and variability from 10 days before to 10 days after OGWD events over the Himalayas. The second subsection shows the

evolution of EP fluxes after the events, which is supposed to explain the dynamical mechanism behind the vortex evolution shown in the previous subsection. The third subsection presents the evolution of total column ozone after the events, and this evolution is tried to be explained by changes in quasi-isentropic mixing. And the last subsection goes back to the dynamical mechanism behind the connection between OGWD events over the Himalayas and the propagation of planetary waves.

A more logical structure would start by showing how OGWD over the Himalayas may change the propagation of planetary waves in the stratosphere. If this is demonstrated, then the changes in the vortex structure and variability, and potential impacts on ozone concentrations can be shown.

Thanks for the suggestion. We revise the manuscript accordingly.

Other comments:

- Figure 1. The evolution of the vortex geometry parameters are not compared to the mean seasonal evolution, and hence we cannot tell if the evolution is statistically different from the mean seasonal evolution. Showing the 95% confidence intervals does not give any useful information in this respect unless the composites show deseasonalized anomalies instead of the global value.

We addressed this comment further above.

- The interpretation of positive anomalies of EPFD as a "wave source" is not correct. When there is a source of planetary waves, there will be divergence of the EP flux (the total field, not the anomaly). There can be a positive anomaly of EPFD but with a total negative divergence. So there is not enough information in Fig. 4 to conclude that there is an anomalous wave source (i.e. it could be weaker convergence).

Besides, a positive EPFD in a total sense in daily data does not directly imply a wave source. Indeed, a propagating wave will induce a flux convergence when arriving at a specific region, and a flux divergence when leaving. So it is important to analyze the time sequence of EP fluxes and divergence to correctly interpret the results.

We checked the absolute values of EPFD above the tropopause and found the positive EPFD anomaly south of the hotspot at 100 hPa as mentioned in the manuscript. We concluded that , together with the time sequence of EP fluxes, it indicates an increased sourcing of PWs with zonal wavenumber 2 above the tropopause (see Fig. below showing anomalies of EPFD (shading) and absolute values (contours) for all wave components). We also revised our statements according to the comment above.

[Figure]

- Please motivate/justify the use of a specified-dynamics climate model simulation. Although the evolution of the nudged variables should be similar to the reanalysis, it is known that other dynamical features such as the residual circulation are not well constrained by the nudging process (Chrysanthou et al. 2019). All reanalyses provide parameterized gravity wave drag output, and ERA5 resolves a good part of the GW spectrum, so I see no reason for going to nudged runs instead of reanalysis.

On the other hand, one of the strengths of using a climate model is the possibility of working with large sample sizes as compared to reanalyses, improving the statistics. But this possibility cannot be exploited using nudged runs.

The fact that CMAM-sd is nudged ensures that the meteorological situation is close to reality, particularly in the troposphere and lower stratosphere to drive OGW parameterization with realistic conditions. On the other hand, we admit that nudging the model dynamics also complicates the causal attribution of composite anomalies due to the two-way interplay between the OGW forcing and the circulation in the middle atmosphere. However, we assessed if nudging has a dynamical impact on the composites in Fig. S1 in Sacha et al (2021). Except locally in the polar mesosphere, the nudging strength composite for the Himalayas is largely not significant and generally smaller than the EPFD or OGWD anomalies allowing us to neglect the nudging effects on the presented results. While CMAM30-SD may be considered an outdated model dataset, currently there is no alternative dataset available with sufficient outputs (3D GWD with daily temporal resolution) to our knowledge, and also not considered in DynVarMIP (Gerber and Manzini, 2016). Therefore, we suggested (see l297-304) that zonally averaged OGWD may be investigated in a similar way using long CMIP6 simulations.

The use of ERA5 would be definitely desirable as we state in the manuscript. However, the assessment of resolved GW spectrum requires methodology on how to extract GWD locally. This kind of methodology is about to be used but as a zonally averaged quantity (Gupta et al, 2021). We convey a finding that resolved and parameterized GWD in ERA5 follows OGWD in CMAM in zonal mean even though with a lower amplitude (see figure below or discussion in Kuchar et al (2020)). Together with former Fig. S2 (current Fig. S3), it suggests that CMAM30-sd simulates the response realistically.

[Figure]

**- Please indicate the number of GWD events identified in the model run.**

The number of days with detected peak events by the peak detection algorithm is shown in Table 1 in Kuchar et al (2020). In addition, Fig. S4 in Sacha et al (2021) suggests that the QBO conditions may play a role as well. Nevertheless, we indicate the number of detected OGWD events (i.e. 37) in the revised manuscript (see l109).

**References**

Butchart, N. (2014). The Brewer-Dobson circulation. *Reviews of Geophysics*, *52*(2), 157–184. https://doi.org/10.1002/2013RG000448

Chrysanthou, A., Maycock, A. C., Chipperfield, M. P., Dhomse, S., Garny, H., Kinnison, D., … Yamashita, Y. (2019). The effect of atmospheric nudging on the stratospheric residual circulation in chemistry–climate models. *Atmospheric Chemistry and Physics*, *19*(17), 11559–11586. https://doi.org/10.5194/acp-19-11559-2019

Edmon, H. J., Hoskins, B. J., & McIntyre, M. E. (1980). Eliassen-Palm Cross Sections for the Troposphere. *Journal of the Atmospheric Sciences*, *37*(12), 2600–2616.

Gerber, E. P. and Manzini, E.: The Dynamics and Variability Model Intercomparison Project (DynVarMIP) for CMIP6: assessing the stratosphere-troposphere system, Geosci. Model Dev., 9, 3413– 3425, https://doi.org/10.5194/gmd-9-3413-2016, 2016

Gupta, A., Birner, T., Dörnbrack, A., & Polichtchouk, I. (2021). Importance of gravity wave forcing for springtime southern polar vortex breakdown as revealed by ERA5. *Geophysical Research Letters*, 48, e2021GL092762. https://doi.org/10.1029/2021GL092762

Jucker, M. (2021). Scaling of Eliassen-   Palm flux vectors. *Atmospheric Science Letters*, *22*(4). https://doi.org/10.1002/asl.1020

Kuchar, A., Sacha, P., Eichinger, R., Jacobi, C., Pisoft, P., & Rieder, H. E. (2020). On the intermittency of orographic gravity wave hotspots and its importance for middle atmosphere dynamics. Weather and Climate Dynamics, 1(2), 481–495. https://doi.org/10.5194/wcd-1-481-2020

Kruse, C. G., Smith, R. B., & Eckermann, S. D. (2016). The Midlatitude Lower-Stratospheric Mountain Wave "Valve Layer." *Journal of the Atmospheric Sciences*, *73*(12), 5081–5100. https://doi.org/10.1175/JAS-D-16-0173.1

Mitchell, D. M., Charlton-Perez, A. J., and Gray, L. J.: Characterizing the Variability and Extremes of the Stratospheric Polar Vortices Using 2D Moment Analysis, Journal of the Atmospheric Sciences, 68, 1194 – 1213, https://doi.org/10.1175/2010JAS3555.1, 2011

Plumb, R. A. (2007), Tracer interrelationships in the stratosphere, *Rev. Geophys.*, 45, RG4005, doi:10.1029/2005RG000179.

Sacha, P., Kuchar, A., Eichinger, R., Pisoft, P., Jacobi, C., & Rieder, H. E. (2021). Diverse Dynamical Response to Orographic Gravity Wave Drag Hotspots—A Zonal Mean Perspective. Geophysical Research Letters, 48(13), 1–11. https://doi.org/10.1029/2021GL093305

Seviour, W. J. M., Mitchell, D. M., & Gray, L. J. (2013). A practical method to identify displaced and split stratospheric polar vortex events. Geophysical Research Letters, 40(19), 5268–5273. https://doi.org/10.1002/grl.50927